# Multidrug-Resistant Bacteria in a COVID-19 Hospital in Zagreb

**DOI:** 10.3390/pathogens12010117

**Published:** 2023-01-10

**Authors:** Branka Bedenić, Vesna Bratić, Slobodan Mihaljević, Anita Lukić, Karlo Vidović, Krešimir Reiner, Silvia Schöenthaler, Ivan Barišić, Gernot Zarfel, Andrea Grisold

**Affiliations:** 1Department for Microbiology and Parasitology, School of Medicine, University of Zagreb, 10000 Zagreb, Croatia; 2Clinical Department for Clinical and Molecular Microbiology, School of Medicine, University Hospital Center Zagreb, University of Zagreb, 10000 Zagreb, Croatia; 3Clinical Department for Anestesiology, Reanimatology and Intensive Care, University Hospital Centre Zagreb, University of Zagreb, 10000 Zagreb, Croatia; 4Varaždin General Hospital, 42000 Varaždin, Croatia; 5Austrian Institute of Technology, 1210 Vienna, Austria; 6Institute for Hygiene, Microbiology and Environmental Medicine, Medical University Graz, 8010 Graz, Austria

**Keywords:** COVID-19, multidrug-resistant bacteria, *Acinetobacter baumannii*, OXA-48

## Abstract

During November to December 2020, a high rate of COVID-19-associated pneumonia with bacterial superinfections due to multidrug-resistant (MDR) pathogens was recorded in a COVID-19 hospital in Zagreb. This study analyzed the causative agents of bacterial superinfections among patients with serious forms of COVID-19. In total, 118 patients were hospitalized in the intensive care unit (ICU) of the COVID-19 hospital. Forty-six out of 118 patients (39%) developed serious bacterial infection (VAP or BSI or both) during their stay in ICU. The total mortality rate was 83/118 (70%). The mortality rate due to bacterial infection or a combination of ARDS with bacterial superinfection was 33% (40/118). Six patients had MDR organisms and 34 had XDR (extensively drug-resistant). The dominant species was *Acinetobacter baumannii* with all isolates (34) being carbapenem-resistant (CRAB) and positive for carbapenem-hydrolyzing oxacillinases (CHDL). One *Escherichia coli* causing pneumonia harboured the *bla*_CTX-M-15_ gene. It appears that the dominant resistance determinants of causative agents depend on the local epidemiology in the particular COVID center. *Acinetobacter baumannii* seems to easily spread in overcrowded ICUs. Croatia belongs to the 15 countries in the world with the highest mortality rate among COVID-19 patients, which could be in part attributable to the high prevalence of bacterial infections in local ICUs.

## 1. Introduction

Secondary bacterial infections such as ventilator-associated pneumoniae (VAP) or bloodstream infections (BSI), particularly with resistant bacteria, seem to complicate clinical presentation of COVID-19 and cause increased mortality and length of hospital stay [1,2]. ESKAPE (*Enterococcus faecium*, *Staphylococcus aureus*, *Klebsiella pneumoniae*, *Acinetobacter baumannii*, *Pseudomonas aeruginosa*, and *Enterobacter* spp.) pathogens are the most frequent isolates associated with VAP and BSI in COVID wards [1].

Frequent multidrug-resistant (MDR) bacteria causing treatment failures are extended-spectrum β-lactamase (ESBL) and/or plasmid-mediated AmpC β-lactamase (p-AmpC) positive Enterobacterales, carbapenem-resistant Enterobacterales (CRE), carbapenem-resistant *Acinetobacter baumannii* (CRAB), carbapenem-resistant *Pseudomonas aeruginosa* (CRPA), methicillin-resistant *Staphylococcus aureus* (MRSA)*,* penicillin-resistant *Streptococcus pneumoniae*, and vancomycin-resistant *Enterococcus* spp. (VRE). ESBLs hydrolyze penicillins, expanded-spectrum cephalosporins (ESC) and monobactams. They belong predominantly to three major families: TEM, SHV and CTX-M, with rare types such as VEB, PER and IBC mainly concentrated in some geographic regions [3]. Plasmids encoding ESBLs often carry resistance genes for non-β-lactam antibiotics such as aminoglycosides, tetracyclines, sulphonamides, chloramphenicol, and fluoroquinolones. The CTX-M family is now dominant with CTX-M-15 allelic variant spreading all over the world [4,5]. 

AmpC β-lactamases are primarily cephalosporinases encoded by chromosomes or plasmids (p-AmpC). P-AmpC β-lactamases are derived from the chromosomally-encoded enzymes of organisms such as *Enterobacter cloacae*, *Citrobacter freundii* or *Morganella morganii*. These enzymes have been detected in *Escherichia coli*, *K. pneumoniae*, *Salmonella* spp., and *Proteus mirabilis* [6]. Those β-lactamases confer resistance to first, second and third generations of cephalosporins, monobactams, and to β-lactam/β-lactamase inhibitor combinations. β-lactamase-mediated resistance to carbapenems in *Enterobacterales* is mostly due to the expression of carbapenemases of class A (KPC), class B (metallo-β-lactamases or MBLs of IMP VIM or NDM series) or class D (OXA-48) [7]. 

Carbapenemases found in CRAB belong to molecular class A (KPC, GES), class B (IMP, VIM, SIM or NDM family) or class D (OXA enzymes) known as CHDL (carbapenem-hydrolyzing class D oxacillinases) [8,9]. Decreased permeability and efflux pump overexpression contribute to carbapenem resistance [8].

MRSA occurs as the result of the acquisition of the *mecA*/*mecC* gene that encodes a novel PBP2a protein. Expression of PBP2a renders bacteria resistant to all β-lactams including cephalosporins (with the exception of ceftaroline and ceftobiprole) [10].

The aim of the study was to analyze MDR and XDR organisms associated with severe COVID-19 secondary bacterial infections, their resistance determinants, and molecular epidemiology.

## 2. Materials and Methods

### 2.1. Patient Data

In total, 118 patients were hospitalized in the COVID-19 ICU, from 1 November until 31 December 2021, in Dubrava Hospital, with a diagnosis of bilateral pneumonia and had positive RT-PCR tests for SARS-CoV-2. The patients were either on high flow oxygen or mechanically ventilated. Dubrava Hospital is the largest center in Croatia for severely ill COVID-19 patients.

The age of the patients and the presence of cardiovascular, kidney and malignant diseases, obesity and diabetes mellitus were recorded from each patient’s medical records in order to determine the risk factors for acquisition of MDR strains. The effect of acquisition of a resistant strain on the duration of hospital stay, the risk of VAP and acute respiratory distress syndrome (ARDS) development and mortality was analysed using statistical methods. VAP was clinically diagnosed in mechanically-ventilated patients by clinical parameters (raised body temperature, chest wall auscultation, blood oxygen saturation), laboratory results (elevated polymorphonuclear leukocyte count, C-reactive protein), and radiological results (chest X-ray, computed tomography), and confirmed by one positive microbiological sample of tracheal aspirate (≥10^5^ CFU) [11]. Endotracheal aspirates (ETA) were collected by trained anesthesiologist. Samples were plated on blood agar and MacConcey agar for semiquantitative culture. Only bacteria numbering >10^4^ were further analyzed. Growth of any microorganism below the threshold was considered as contamination or colonization. VAP was defined as having either an early- or a late-onset according to whether it began before or after the first 5 days of hospitalization [11].

A bloodstream infection (BSI) was diagnosed based on the presence of causative agents in the blood accompanied by SIRS or systemic inflammatory response (elevated or decreased temperature (>38.5 °C or <36 °C), increased heart rate (>90), respiratory rate (>20) and white blood cell count (≥12 × 10^9^/L), or leukopenia (≤10^4^/L) [12]. Blood culture (BC) bottles were used for routine cultivation and diagnosis, which included BACTEC FX (BioMerieux, Marcy, l’Etoile, France). Positive BCs were subjected to Gram staining and subcultured on solid medium (blood, chocolate and Columbia agar), and after 18 to 24 h incubation (overnight), were identified by MALDI-TOF MS (matrix-assisted laser desorption ionization-time of flight mass spectrometry) Biotyper (Bruker, Daltonik GmbH, Bremen, Germany) [13]. One or two colonies of each isolate were directly spotted on the manufacturer’s proprietary sample plates following the manufacturer’s protocols and recommendations. A 1-µL volume of CHCA matrix solution (α-cyano-4-hydroxycinnamic acid; BioMérieux Inc., Marcy-l’Étoile, France) was then applied to each sample and air-dried for 5 min at room temperature for crystallization. For species identification of each isolate, a total of four spots were analyzed on the VITEK MS system (BioMérieux Inc., Marcy-l’Étoile France). The MALDI-TOF MS instrument used in this study was equipped with a 337 nm-fixed focus nitrogen laser of 50 Hz frequency, the software program was the VITEK MS IVD analysis software version 3.2. *Escherichia coli* ATCC 8739 was used as positive control.

### 2.2. Bacterial Isolates

Non-copy bacterial isolates (one per patient) were recovered from various clinical samples including clinically relevant (BCs cultures and ETAs), urine samples, and surveillance cultures (rectal swabs, throat swab, etc.). 

### 2.3. Antimicrobial Susceptibility Testing

The antimicrobial susceptibility of Gram-negative bacteria to a wide range of antibiotics was determined by the broth microdilution method according to CLSI standards [14] and for colistin according to the EUCAST standards [15]. The panel of antibiotics depended on the species as shown in Table 1, Table 2, Table 3 and Table 4. Susceptibility of Gram-positive bacteria was tested by routine disk-diffusion method except for vancomycin and teicoplanin, for which a dilution test was carried out.

### 2.4. Phenotypic Detection of β-Lactamases

ESBLs were detected by a double disk synergy test [16] and combined disk test with cephalosporins and clavulanic acid [14]. For *A. baumannii*, the test was carried out with the addition of cloxacillin in the medium (200 mg/L) to inhibit the chromosomal AmpC β-lactamase, which can antagonize the synergistic effect with clavulanate [17].

Plasmid-mediated AmpC β-lactamases were detected in *K. pneumoniae* and *E. coli* by combined disk test using cephalosporin disks combined with 3-aminophenylboronic acid (PBA) [18]. A modified Hodge test (MHT) with imipenem disk was used to screen for the production of carbapenemases [19]. Additionally, the isolates were tested by combined disk tests with imipenem and meropenem alone and combined with PBA, 0.1 M EDTA, or both to screen for KPC, MBLs, or simultaneous production of KPC and MBL, respectively [20]. To confirm carbapenem hydrolysis in carbapenem-resistant isolates, a CIM (carbapenem inactivation method) test was performed [21]. A suspension of the test strain was adjusted to McFarland 0.5 (10^8^ CFU/mL) and a meropenem disk was placed in the suspension. The suspension was incubated for 2 h at 37 °C. *E. coli* ATCC 25922 was inoculated on Mueller–Hinton agar (MH) and the disk was placed in the middle of the plate. The plates were incubated overnight and the lack of inhibition zone or colonies within the inhibition zone indicated carbapenem hydrolysis. 

The isolates were classified as multidrug-resistant (MDR), extensively-drug-resistant (XDR) and pandrug-resistant (PDR) as described previously by Magiorakos et al. [22]. 

### 2.5. Conjugation

The transferability of cefotaxime or ertapenem resistance was determined in *Enterobacterales* by conjugation (broth mating method) employing *E. coli* J65 resistant to sodium azide [23]. The ESBL- and carbapenemase-producing transconjugants were selected on MacConkey agar containing either ertapenem (0.5 mg/L) or cefotaxime (2 mg/L) and sodium azide (100 mg/L). The frequency of conjugation was determined relative to the number of donor cells. Cotransfer of resistance to gentamicin, tetracycline, sulfamethoxazole/trimethoprim, chloramphenicol and ciprofloxacin was also determined. 

### 2.6. Molecular Detection of Resistance Genes

The nature of ESBL, carbapenemases and fluoroquinolone resistance determinants was investigated by PCR. The genes conferring resistance to β-lactams including broad-spectrum and extended-spectrum β-lactamases (*bla*_SHV_, *bla*_TEM_, *bla*_CTX-M_, and *bla*_PER-1_) [24,25,26,27], plasmid-mediated AmpC β-lactamases (p-AmpC) [28], class A (*bla*_KPC_), class B or metallo-β-lactamases-MBLs (*bla*_VIM_, *bla*_IMP_, and *bla*_NDM_) and class D carbapenemases or carbapenem-hydrolyzing oxacillinases-CHDL (*bla*_OXA-48_) [29] and to fluoroquinolones (*qnr*A, *qnr*B, and *qnr*S) [30] were sought in *Enterobacterales*. Multiplex PCR was applied to determine the five clonal lineage of CTX-M β-lactamases [31]. Genetic context of *bla*_CTX-M_ genes was determined by PCR mapping with forward primer for IS*Ecp1* and IS*26* combined with primer MA-3 (reverse for *bla*_CTX-M_ genes) [32].

In *A. baumannii* isolates, genes encoding KPC, MBLs (*bla*_VIM_, *bla*_IMP_, *bla*_SIM_ and *bla*_NDM_) and CHDL *(bla*_OXA-51-like_, *bla*_OXA-23-like_, *bla*_OXA-24/40-like_, *bla*_OXA-58-like_, and *bla*_OXA-143-like_) were determined by PCR using protocols and conditions as described previously [29,33]. The genetic context of *bla*_OXA-51_ and *bl*a_OXA-23_ genes in *A. baumannii* was determined by PCR mapping with primers for IS*Aba1* combined with forward and reverse primers for *bla*_OXA-51_ and *bla*_OXA-23_ according to Turton et al. [34].

### 2.7. Whole Genome Sequencing (WGS)

Nine randomly-selected *A. baumannii* isolates were subjected to WGS. First, the strains were cultivated in Tryptic Soy Broth (TSB) and Casein-Peptone Soymeal-Peptone (CASO) Broth (Merck Millipore, MA, USA) at 37 °C overnight. Then, the genomic DNA was extracted using the QIAamp UCP Pathogen Mini Kit (Qiagen, Hilden, Germany) according to the manufacturer’s instructions. The DNA extracts were sent to the Next Generation Sequencing Facility of the Vienna Biocenter for sequencing using Illumina’s NextSeq1000 system according to the manufacturer’s instructions. The single reads obtained were assembled and analysed using the webservers and services of the Center for Genomic Epidemiology (http://www.genomicepidemiology.org (accessed on 13 January 2022) [35]. The sequences were deposited in the NCBI Gen Bank, and the accession numbers were provided. 

### 2.8. Characterization of Plasmids and Molecular Typing of A. baumannii Isolates

Plasmids were extracted with a Qiagen Mini kit (Inel, Croatia) according to the manufacturer’s instructions. PCR-based replicon typing (PBRT) was according to Carattoli et al. [36]. Since it was observed previously that PBRT can be inefficient in identifying L/M plasmid incompatibility type, an updated method designated to identify and distinguish between IncL and IncM plasmids was applied [37]. PBRT according to Bertini was applied for *A. baumannii* to type the resistance plasmids carrying carbapenemase genes [38]. 

Sequence groups (SGs 1–3) corresponding to international clonal lineages (ICL I-III) determination was performed according to the procedure described by Turton et al. [39]. Six selected isolates (number 3, 4, 5, 7, 9, and 10 in Table 2) were subjected to multilocus sequence typing (MLST) in order to determine the sequence types (ST) according to the Pasteur website (Multilocus sequence typing (MLST) databases and software for *A. baumannii*. Seven housekeeping genes were amplified and the PCR products were detected by agarose gel electrophoresis, purified and sequenced using the Eurofin service. The obtained sequences were deposited into the above-mentioned website in order to obtain the ST. (Available online: https://pubmlst.org/abaumannii/, accessed on 13 January 2022).

### 2.9. Statistical Analysis

The data, describing demographics, number of ventilator hours, as well as number of ICU days, were summarized by the means of descriptive statistics (median, range, interquartile range due to the abnormal distribution, D’Agostino–Pearson test). Continuous variables were compared using the Mann–Whitney test for independent samples, while categorical variables were compared using a χ-square test for independent samples. All statistical analyses were performed using MedCalc version 9.5.1.0 statistical software (MedCalc Software, Mariakerke, Belgium). *p* values < 0.05 were considered as statistically significant. The relationship between severity of infection (severe: BSI, VAP, or combined) or presence of MDR/XDR bacteria (compared to sensitive bacteria) and outcome of the patient was investigated using logistic regression.

## 3. Results

### 3.1. Patients

In total there were 118 patients in the study (78 males and 40 females). The median age was 71 years (range 25–94). Eighty-three patients (70%) with bilateral pneumonia developed ARDS. The average duration of stay in ICU was 7.4 days (range 0–61). The total mortality rate was 83/118 (70%)

Forty-six out of 118 patients (39%) developed serious bacterial infection (VAP or BSI or both) during their stay in the ICU. VAP (late onset) was diagnosed in 33 patients (28%). Twenty-seven patients (23%) had BSI. Fourteen patients (12%) had simultaneous VAP and BSI. All patients with serious infections had elevated white blood cells count, CRP and X-ray results consistent with severe bilateral pneumonia. For the clinicians, it was difficult to distinguish if the deterioration was due to secondary bacterial infection or the progression of COVID-19 but was most probably due to the combination of ARDS and secondary bacterial infections, leading to multiorgan failure (MOF). The mortality rate due to bacterial infection or the combination of ARDS with bacterial superinfection was 33% (40/118). In total, 83 patients died. Thus, 40 death cases (48%) were attributed to bacterial infection, complicating COVID-19 disease, while the remaining 43 patients (52%) died from the progression of COVID-19 disease. The lethal outcome was recorded in 33/40 patients (80%) having serious bacterial infection due to MDR or XDR organisms and in seven out of eight patients (87%) with susceptible organisms associated with BSI. Less serious infections diagnosed in patients were urinary tract infections (UTI), diagnosed in only eight (6.7%) patients, and pre-existing wound infections identified in three (2.5%) of the patients. Age and gender were not found to be a risk factor for acquisition of MDR or XDR isolates with *P* values of 0.7 and 0.4, respectively. Comorbidities such as diabetes mellitus, asthma, obesity/overweight, polyarthritis, chronic kidney failure, and malignant diseases were not identified as risk factors for infections with resistant isolates with *p* values of 0.4, 0.11, 1.00, 0.61, 0.73 and 0.65, respectively. A significant *p* value was obtained only for cardiovascular diseases (*p* = 0.03). A logistic regression showed a strong positive relationship between the presence of serious infection and mortality (*p* = 0.001, odds ratio 4.341 with 95% CI 1.627 to 11.581, AUC 0.653 with 95% CI 0.559 to 0.738). On the other hand, it showed no relationship between the presence of the XDR/MDR bacterial strain as the causative agent of the serious infection and mortality (*p* = 0.960, odds ratio 1.061 with 95% CI 0.107 to 10.544, AUC 0.504 with 95% CI 0.353 to 0.655).

### 3.2. Bacterial Isolates

In total, there were 48 isolates from clinically-relevant specimens (ETA and BSI). There were 40 resistant isolates and eight susceptible. Thirty-four patients had *A. baumannii*, eight had methicillin-susceptible *Staphylococcus* spp., five had MRSA, and one had *E. coli*. Thirty-four patients had XDR organisms (CRAB) and six had MDR (one *E. coli* ESBL and five MRSA) associated with severe infection. *Staphylococcus* spp. identified in the BC of eight patients were susceptible. 

Twenty-nine patients with VAP had XDR organism (CRAB) and five MDR (one ESBL positive *E. coli* and four MRSA). One patient had MRSA and *A. baumannii* in the same specimen. CRAB (XDR) was isolated from 18 patients and MRSA (MDR) from one patient with BSI, while eight BCs grew methicillin-susceptible organisms (three *Staphylococcus epidermidis*, and one *Staphylococcus aureus*, *Staphylococcus haemolyticus*, *Staphylococcus pettenkoferi*, and *Staphylococcus hominis*).

UTIs were mostly due to resistant Enterobacterales (two OXA-48 positive *K. pneumoniae*, one ESBL-positive *K. pneumoniae* and *E. coli*, respectively, and one MRSA) while three patients developed UTI caused by susceptible bacteria (one *K. oxytoca*, *Proteus mirabilis* and *Enterobacter cloacae* each). Two patients had wound infections with MDR organism: one OXA-48 positive *K. pneumoniae* and VRE, respectively, whereas one had a drain infection with susceptible *P. aeruginosa*. Colonization with an MDR organism was recorded in surveillance cultures (throat swab, rectum swab, or stool) of three patients (one *K. pneumoniae* OXA-48, *E. coli* ESBL and VRE). All patients with invasive infections due to *A. baumannii* had also the same strain with identical resistance patterns in surveillance cultures (nasopharyngeal swab, throat swab, axilla swab, stool, or rectum swab). The non-invasive isolates were not further analysed. Four patients’ urinary tracts were colonized with *Candida* spp. The results pertaining to UTI isolates and surveillance cultures are shown in the Appendix A. 

### 3.3. Antibiotic Susceptibility

Forty non-copy resistant bacterial isolates (one per patient) were recovered from clinically relevant specimens: BCs and ETAs. Among clinically relevant resistant isolates there were 34 *A. baumannii* (XDR), one *E. coli*, and five MRSA (MDR). Eight susceptible *Staphylococcus* spp isolates were identified in BCs and they were not further analysed. Two urinary *K. pneumoniae* isolates were XDR, whereas two *Enterobacterales* and one MRSA were MDR, and three *Enterobacterales* S as shown in the Appendix A. 

#### 3.3.1. Enterobacterales

There was only one ESBL-positive *E.coli* isolate from ETA. The isolate exhibited resistance to ESC, gentamicin, and ciprofloxacin, and susceptibility to carbapenems. Results are shown in Table 1. The remaining seven resistant isolates originated from urine or surveillance cultures (five *K. pneumoniae* and two *E. coli*) and are presented in the Appendix A. 

**Table 1 pathogens-12-00117-t001:** Minimum inhibitory concentrations, susceptibiltiy category, phenotypic tests for beta-lactamase detection and β-lactamase content of Enterobacterales isolates. MICs were interpreted according to CLSI standards, except for colistin, which was carried out according to EUCAST guidelines.

Strain	SPECIMEN and Outcome	ESBL	AMX	AMC	TZP	CAZ	CTX	CRO	FEP	IPM	MEM	ERT	GM	CIP	COL	BL
*E. coli* 308523	ETA D	+	>128	64	64	>128	>128	>128	64	1	0.5	S	32	64	0.25	CTX-M-15

Abbreviations: AMX—amoxycillin; AMC—amoxycillin/clavulanic acid; TZP—piperacillin/tazobactam; CAZ—ceftazidime; CTX—cefotaxime; CRO— ceftriaxone; FEP—cefepime; IMI—imipenem; MEM—meropenem; ERT—ertapenem; GM—gentamicin; CIP—ciprofloxacin; COL—colistin; ESBL—inhibitor based test with clavulanic acid for detection of extended-spectrum beta-lactamases; BL—beta-lactamase content; D—death.

#### 3.3.2. *Acinetobacter baumannii*

Only isolates from clinically-relevant specimens (ETA and BC) were subjected to laboratory analysis. *A. baumannii* isolates were uniformly resistant to piperacillin/tazobactam, ceftazidime, cefepime, imipenem, meropenem, gentamicin, and ciprofloxacin, but uniformly susceptible to colistin (Table 2). Sulbactam/ampicillin exhibited good activity with 74% of the isolates being susceptible (n = 25). The MICs of carbapenems were not lowered by cloxacillin, indicating that hyperproduction of chromosomal AmpC β-lactamase did not contribute to carbapenem resistance. CIM and Hodge tests were positive in all *A. baumannii*, indicating carbapenemase production (Table 2). 

**Table 2 pathogens-12-00117-t002:** Antibiotic susceptibility, phenotypic tests for detection of β-lactamases and *bla*_OXA_ gene content of *A. baumanii* isolates. MICs were interpreted according to CLSI standards, except for colistin which was carried out according to EUCAST guidelines.

	POTOCOL NUMBER	OUTCOME	SPECIMEN	Hodge	CIM	EDTA	TZP	CAZ	FEP	IMI	MEM	GM	CIP	SAM	COL	IC	BL, ST
1	3791, 1998	S	BC, ETA	+	+	+	>128	>128	>128	>128	>128	>128	>128	8	1	2	OXA-72
2	7548	S	BC	+	+	+	>128	>128	>128	>128	>128	>128	>128	4	1	2	OXA-23
3	305574	D	BC	+	+	+	>128	>128	>128	>128	>128	>128	>128	4	1	2	OXA-23, ST208
4	299055	D	ETA	+	+	+	>128	>128	>128	>128	>128	>128	>128	8	1	2	OXA-23, ST425
5	314959	D	BC	+	+	+	>128	>128	>128	>128	>128	>128	>128	8	2	2	OXA-23, ST195
6	317893	D	ETA	+	+	+	>128	>128	>128	>128	>128	>128	>128	8	2	2	OXA-23
7	297466	D	ETA	+	+	+	>128	>128	>128	>128	>128	>128	>128	16	2	2	OXA-23, ST748
8	8959	D	ETA	+	+	+	>128	>128	>128	>128	>128	>128	>128	16	2	2	OXA-72
9	317063	S	BC	+	+	+	>128	>128	>128	>128	>128	>128	>128	4	1	2	OXA-23, ST478
10	290040	D	ETA	+	+	+	>128	>128	>128	>128	>128	>128	>128	32	1	2	OXA-72,ST208
11	288237 294604	D	BC, ETA	+	+	+	>128	>128	>128	>128	>128	>128	>128	32	0.5	2	OXA-23-like
12	310639	D	BC, ETA	+	+	+	>128	>128	>128	>128	>128	>128	>128	16	2	2	OXA-24-like
13	290005	D	ETA	+	+	+	>128	>128	>128	>128	>128	>128	>128	16	1	2	OXA-24-like
14	4853	D	BC	+	+	+	>128	>128	>128	>128	>128	>128	>128	4	1	2	OXA-24-like
15	295429 295432	D	BC, ETA	+	+	+	>128	>128	>128	>128	>128	>128	>128	4	1	2	OXA-24-like
16	4829, 4841	D	BC	+	+	+	>128	>128	>128	>128	>128	>128	>128	4	1	2	OXA-24-like
17	8249	D	ETA	+	+	+	>128	>128	>128	>128	>128	>128	>128	32	1	2	OXA-24-like
18	294599 290006	D	BC, ETA	+	+	+	>128	>128	>128	>128	>128	>128	>128	64	1	2	OXA-24-like
19	316223	D	ETA	+	+	+	>128	>128	>128	>128	>128	>128	>128	32	0.25	2	OXA-24-like
20	300705	D	BC, ETA	+	+	+	>128	>128	>128	>128	>128	>128	>128	4	1	2	OXA-24-like
21	309328	D	BC, ETA	+	+	+	>128	>128	>128	>128	>128	>128	>128	4	0.5	2	OXA-24-like
22	289675	D	ETA	+	+	+	>128	>128	>128	>128	>128	>128	>128	8	0.5	2	OXA-24-like
23	20584	D	ETA	+	+	+	>128	>128	>128	>128	>128	>128	>128	4	0.5	2	OXA-24-like
24	300700	D	ETA	+	+	+	>128	>128	>128	>128	>128	>128	>128	32	0.5	2	OXA-24-like
25	307477	D	BC, ETA	+	+	+	>128	>128	>128	>128	>128	>128	>128	>128	0.5	2	OXA-24-like
26	287935	D	BC	+	+	+	>128	>128	>128	>128	>128	>128	>128	32	0.5	2	OXA-24-like
27	290410	D	BC, ETA	+	+	+	>128	>128	>128	>128	>128	>128	>128	16	0.5	2	OXA-24-like
28	290000	D	ETA	+	+	+	>128	>128	>128	>128	>128	>128	>128	16	1	2	OXA-24-like
29	316358	D	BC, ETA	+	+	+	>128	>128	>128	>128	>128	>128	>128	4	1	2	OXA-24-like
30	288243	D	BC	+	+	+	>128	>128	>128	>128	>128	>128	>128	64	1	2	OXA-24-like
31	297099 297105		BC, ETA	+	+	+	>128	>128	>128	>128	>128	>128	>128	8	1	2	OXA-24-like
32	314854	D	ETA	+	+	+	>128	>128	>128	>128	>128	>128	>128	4	1	2	OXA-23-like
33	306000	D	ETA	+	+	+	>128	>128	>128	>128	>128	>128	>128	4	1	2	OXA-23-like
34	302294	D	ETA	+	+	+	>128	>128	>128	>128	>128	>128	>128	2	1	2	OXA-24-like

Abbreviations: CAZ—ceftazidime; FEP—cefepime; IMI—imipenem; MEM—meropenem; SAM—ampicillin/sulbactam; GM—gentamicin; CIP—ciprofloxacin; COL—colistin; ESBL—inhibitor based test with clavulanic acid for detection of extended-spectrum β-lactamases; BL—β-lactamase content; CIM—carbapenem inactivation method; EDTA—combined disk test for detection of MBLs; BC: blood culture; ETA—endotracheal aspirate; IC: international clonal lineage; ST—sequence type; all isolates harboured intrinsic *bla*_OXA-51_-like gene, D—death, S—survival.

#### 3.3.3. Gram-Positive Isolates

All five MRSA isolates recovered from relevant specimens, were uniformly susceptible to sulphamethoxazole/trimethoprim, vancomycin, teicoplanin, and rifampicin (Table 3.) All were resistant to clindamycin and erythromycin, whereas three isolates exhibited resistance to ciprofloxacin and one to gentamicin. Two VRE isolates were obtained from wound swab and surveillance cultures (stool). The results are shown in the Appendix A.

**Table 3 pathogens-12-00117-t003:** Antibiotic susceptibility of methicillin-resistant *Staphylococcus aures* (MRSA). Disk diffusion test was performed and interpreted according to the CLSI guidelines.

PROTOCOL NUMBER	SPECIMEN and Outcome	PEN	OX	CLY	ERI	SXT	RIF	GM	CIP	VAN	TEIC	LZD
1. (7620)	ETA D	R	R	R	R	S	S	S	R	S (0.5)	S (0.5)	S
2. (317779)	ETA S	R	R	R	R	S	S	S	R	S (0.5)	S (0.5)	S
3. (305541)	ETA D	R	R	R	R	S	S	S	S	S (0.5)	S (0.5)	S
4. (302595)	ETA D	R	R	R	R	S	S	S	S	S (0.5)	S (0.5)	S
5. (288240)	BC D	R	R	R	R	S	S	R	R	S (0.5)	S (0.5)	S

Abbreviations: Pen—penicillin; OX—oxacillin; SXT—sulphamethoxazole/trimethoprim; RIF—rifapicin; GM—gentamicin; CIP—ciprofloxacin; VAN—vancomycin; TEIC—teicoplanin; LZD—linezolid; CLY—clindamycin; ERY—erythromycin; for vancomycin and teicoplanin MIC value is shown; ETA—endotracheal aspirate; BC—blood culture; D—death; S—survival.

### 3.4. Conjugation

Cefotaxime resistance transfer was not successful from the ESBL-positive *E. coli* isolate.

### 3.5. Molecular Detection of Resistance Genes

#### 3.5.1. Enterobacterales

*E. coli* yielded product only with primers specific for *bla*_CTX-M-15_ genes (Table 1). IS*Ecp* was identified upstream of *bla*_CTX-M-15_ gene. 

#### 3.5.2. *A. baumannii*

*bla*_OXA-24-like_ and *bla*_OXA-23-like_ genes were identified in *A. baumannii* isolates in 24 and 10 isolates, respectively, as shown in Table 3. *bla*_OXA-23-like_ genes and *bla*_OXA-51-like_ genes were preceded by IS*Aba1*.

### 3.6. Whole Genome Sequencing

All nine tested *A. baumannii* isolates harboured aminoglycoside-resistant genes: *arm*A encoding 16S rRNA methyltransferase providing panaminoglycoside resistance and *aph*(3″)-Ib and *aph*(6)-Id for aminoglycoside phosphotransferases as shown in Table 4. *Sul*1 and *sul*2 genes, responsible for sulphonamide resistance and *tet*B associated with tetracycline resistance were identified. Among β-lactam resistance determinants, *bla*_ADC-25_ encoding chromosomal cephalosporinase and intrinsic, chromosomal *bla*_OXA-66_ were detected in all isolates. Acquired *bla*_OXA-23_ and *bla*_OXA-72_ were found in seven and three isolates, respectively, as shown in Table 4. Accession numbers are provided in Table 4.

**Table 4 pathogens-12-00117-t004:** Whole genome sequencing of *Acinetobacter baumannii* isolates.

Isolate	PROTOCOL NUMBER	AG	β-Lactam	SUL	TET	CHL	ACCCESION NUMBER
*A. baumannii* 1	3791	*arm*A		*Sul*2	*tetB*		JAKLXV000000000
		*aph*(3″)*-Ib* *aph*(6)-*Id*	*bla* _OXA-66_ *bla* _ADC-25_ *bla* _OXA-72_				
*A. baumannii* 2	7548			*sul*1	*tetB*		JAKLXW000000000
		*arm*A	*bla* _OXA-66_ *bla* _ADC-25_ *bla* _OXA-23_				
		*aph*(3″)*-Ib* *aph*(6)-*Id*					
	305,574	*arm*A	*bla* _OXA-66_ *bla* _ADC-25_ *bla* _OXA-23_	*sul*1	*tetB*		JAKLXX000000000
*A. baumannii* 3		*aph*(3″)*-Ib* *aph*(6)-*Id*					
	314,959	*arm*A	*bla* _OXA-66_ *bla* _ADC-25_ *bla* _OXA-23_	*sul*1	*tetB*		JAKLXY000000000
*A. baumannii* 5		*aph*(3″)*-Ib* *aph*(6)-*Id*					
	317,893						JAKLXZ000000000
*A. baumannii* 6		*arm*A	*bla* _OXA-66_ *bla* _ADC-25_ *bla* _OXA-23_	*sul*1	*tetB*		
		*aph*(3″)*-Ib* *aph*(6)-*Id*					
*A. baumannii* 7	297,466	*arm*A	*bla* _OXA-66_ *bla* _ADC-25_ *bla* _OXA-23_	*sul*1	*tetB*	*cat*A1	JAKLYA000000000
		*aph*(3″)*-Ib* *aph*(6)-*Id* *aac*(3)-*Ia* *aadA*1					
*A. baumannii* 8	8959	*arm*A		*Sul*2	*tetB*		JAKLYB000000000
		*aph*(3″)*-Ib* *aph*(6)-*Id* *aac*(3)-*Ia* *aadA*1	*bla* _OXA-66_ *bla* _ADC-25_ *bla* _OXA-72_				
*A. baumannii* 9	317,063	*arm*A	*bla* _OXA-66_ *bla* _ADC-25_ *bla* _OXA-23_	*sul*1	*tetB*		JAKLYC000000000
		*aph*(3″)*-Ib* *aph*(6)-*Id* *aadA*1					
	290,040	*arm*A		*Sul*1	*tetB*		
*A. baumannii* 10		*aph*(3″)*-Ib* *aph*(6)-*Id* *aac*(6)-Ip	*bla* _OXA-66_ *bla* _ADC-25_ *bla* _OXA-72_				JAKLYD000000000

### 3.7. Plasmid Analysis

#### 3.7.1. Enterobacterales 

*E. coli* strain and its transconjugant were positive for IncFIA plasmid.

#### 3.7.2. *A. baumannii*

The plasmids extracted from isolates positive for OXA-23 belonged to Inc group 6 encoding aci6-replicase gene. 

### 3.8. Genotyping

All *A. baumannii* isolates belonged to SG1 corresponding to ICII. Six randomly selected isolates were subjected to MLST. Two isolates belonged to the ST748. The other five ware all members of the Clonal complex 208 (CC208), with two ST208, one ST195 and one ST425. These three STs differ only in their pattern in the *gpi* gene.

## 4. Discussion

The main finding of the study is the high rate of MDR pathogens among patients hospitalized in the COVID-19 hospital in Zagreb. This is in contrast to previous studies reporting low rates of bacterial infections in COVID-19 patients in some COVID centers, but with high use of broad-spectrum antibiotics in empirical therapy [40,41]. The explanation might be the high usage of immunosuppressive therapy in order to prevent ARDS. The rate of MDR pathogens among COVID patients similar to our study was reported in Italy—35% [42]. In Italy, *Enterobacterales* and MRSA were the dominant MDR pathogens associated with bacterial coinfections [42]. However, the mortality rate was markedly lower in the Italian study compared to our results (30% vs. 70%). For the clinicians, it is difficult to clinically differentiate COVID-19 progression from bacterial or fungal superinfection and thus antibiotic therapy is often initiated too late. In our study, lethal outcomes were assigned to the combined effect of ARDS and bacterial infections leading to MOF. Our results are contradict those reported from Spain, where typical community-acquired pathogens such as *Streptococcus pneumoniae* were the dominant causative agents of bacterial superinfections in COVID-19-hospitalized patients, which occurred in only 3% of the patients [43]. Low microbiological sampling and immediate empirical antibiotic therapy upon hospitalization could contribute to low infection rates. Similarly, low rates of bacterial infections (6%) were observed in UK hospitals [44] with *S. aureus* and *P. aeruginosa* as dominant pathogens in late onset VAP, unlike our results. Previous investigations have shown that bacterial coinfections are not directly attributable to SARS-CoV-2, but rather to staying in overcrowded ICUs, lapses in infection control measures, and antibiotic overuse in COVID-19 patients [40]. For those reasons, they are associated with hospital pathogens endemic in local hospitals and ICUs such as CRAB, CRE or MRSA. On the other hand, in Wuhan, where the pandemics started, high rates of secondary infections with MDR isolates were observed, ranging from 5 to 27% with 50% of deceased patients having secondary infections [45,46,47]. On the contrary, in our study the acquisition of MDR isolates was not related to increased mortality. Statistical analysis found no correlation between the susceptibility category and the patient’s outcome. Bacterial infection itself was identified as the risk factor for the lethal outcome. However, resistance traits did not seem to affect the mortality rate. There was a high mortality rate among patients having BSI with typical skin microbiota without relevant virulence determinants. This could be due to the very poor functional status of mechanically-ventilated patients. The high rates of resistant bacteria in ETAs in our study are in accordance with the fact that all VAP cases were late onset and acquired in the hospital. 

An important finding in this study was that CRAB was the dominant causative agent of VAP and BSI in COVID-19 patients, whereas Enterobacterales, particularly OXA-48-positive *K. pneumoniae*, were dominantly associated with UTI or colonization. Similar results were reported during COVID-19 pandemics from Mexico, where clonal spread of CRAB was observed [1]. The isolates analysed in their study were identified in infusion pumps, the oxygen source, vital sign monitors, ultrasound equipment, computer keyboards, and other inanimate surfaces in the hospital. This finding poses a serious challenge for clinicians working in COVID units because the new antibiotics such as ceftazidime/avibactam, ceftolozane/tazobactam and imipenem/cilastatin/relebactam have no activity on CRAB. The resistance determinants found in this study are in line with previously published results from the same geographic area, indicating that OXA-24-like and OXA-23-like are endemic among *A. baumannii* in Zagreb hospitals and in nursing homes also [48,49,50,51]. Recently pandrug-resistant *A. baumannii* isolates harbouring OXA-23 and colistin-resistant determinants were described in a hospital centre in eastern region of Croatia [52], but in the current study all isolates were susceptible to colistin, in spite of the fact that colistin is often used to treat BSI in COVID centers. Colistin resistance in Croatian *A. baumannii* isolates was previously reported to be related to single nucleotide polymorphism in *pmr*B and *mgr*B genes [52]. ST195 was previously identified in *A. baumannii* in Croatia in both clinical and environmental isolates [50,51]. STs 478 and 748 were reported for the first time in Croatia. It seems that the same resistance determinants as in previous studies, were carried by new STs, demonstrating the ability of *A. baumannii* to change the population structure but keep the same CHDL. The dominant MLST clonal complex in this study was CC208, this agrees with its dominant or strong presence in Europe and other parts of the world [53,54]. On the other hand, the results from India identified *K. pneumoniae* as the dominant pathogen in seriously ill COVID-19 patients [2]. The dominant resistance traits found in their isolates were NDM-1 and OXA-48. Similarly, as in our study, VAP and bloodstream infections were the dominant bacterial coinfections in COVID-19 patients, leading to MOF and lethal outcomes. In our study MRSA was the second-most frequent causative agent of VAP but the molecular analysis of resistance traits was not carried out. 

*E. coli* isolate produced CTX-M-15 which is endemic in Croatia and all over the world [55,56]. It conferred on the producing isolates a high level of resistance to expanded-spectrum cephalosporins (ESC). In spite of being positive for IS*Ecp1*, which mobilizes the *bla*_CTX-M_ genes, cefotaxime resistance was not transferable. 

FIIs was found among CTX-M-15 producing *E. coli* isolates from Zagreb [56], indicating that the same plasmid incompatibility groups are circulating in this geographic region during the prolonged period and found their way to the COVID hospital. Laboratory identification of resistance traits is important to track the spread of resistant isolates in COVID hospital wards in order to identify the source and routes of spread. 

The limitation of the study is the inclusion of only one COVID hospital in the study. Since the patients were admitted to the COVID ICU from other hospitals, nursing homes or emergency units, it is not clear if they had resistant isolates before being admitted to the COVID center. There was no time and no staff to perform screening on MDR bacteria upon admission. The limit of the study is that our study might not have the power to show an independent association between MDR and XDR bacterial phenotypes and mortality. The strength of the study is the detailed molecular analysis of resistance genes and the plasmids carrying them. There are many published reports on bacterial pathogens causing coinfections in COVID-19 patients, but molecular analysis of resistant traits is rarely reported. 

The mechanisms predisposing for bacterial superinfections in patients with patients suffering from bilateral COVID-19 pneumonia have to be clarified in future studies. The high level of mortality arising from intrahospital infections with MDR strains, as seen in this study, is reason for alarm. It appears that the lack of control of contamination sources and hygiene caused the dissemination of microorganisms among patients. However, there was no direct evidence of the source of contamination or transmission of the MDR strains, although it was probably due to transitory hand contact through hospital personnel. For this reason, it is important to characterize the acquisition of resistance traits among MDR isolates.

## 5. Conclusions

In conclusion, it appears that the rate of bacterial superinfections and the dominant MDR causative agents depend on the local epidemiology in the particular COVID centers. *A. baumannii* seems to spread in overcrowded ICUs with poor infection control measures and lapses in hospital hygiene measures. Croatia belongs to the 15 countries in the world with the highest mortality rate among COVID-19 patients, which could be in part attributable to high incidence of secondary bacterial infections associated with MOF and poor outcome.

## Data Availability

The data presented in this study are available on request from the corresponding author.

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
