# Peer review of "Multidrug-Resistant Bacteria in a COVID-19 Hospital in Zagreb"

_pathogens, 2023, doi:10.3390/pathogens12010117_

Round 1
Reviewer 1 Report
The manuscript deals with a topic of great relevance, and has adequate scientific merit. Therefore, I consider it suitable for publication after some corrections.
Abstract:
Line 14: change to "multidrug-resistance (MDR)" or "multidrug-resistant (MDR) pathogens"
Line 17: Define ICU and XDR in the abstract before the use of abbreviation.
Introduction:
Lines 31-34: The correlation of COVID with other illnesses was slightly vague in this sentence, being further clarified in the next sentence. I suggest readjusting this excerpt, or removing it, as the content is better described in the next sentence.
Lines 33 and 35: Cite the abbreviation only once.
Line 37: Define ESKAPE before the use of abbreviation.
At the end of the introduction, mention the research objectives.
Material and methods:
Line 75: Define ARDS before the use of abbreviation.
Lines 77-81: Describe whether samples were collected from all patients, but only 49 bacteria were isolated, or if only 49 samples were collected.
Lines 80-81: The order must be inverted, first the full name, then, in parentheses, the abbreviation.
Lines 82-89: The statistics session must be the last of the methods session.
Tables 1-4: Mention in the description of the table that these values are the reference standards. In addition, it is appropriate to indicate the source of the reference used in the legend of the table (CLSI, EUCAST, etc), even if it has already been cited in the text.
Line 123: There is no need for [CLSI] on that line. The CLSI has already been mentioned as a reference of the method previously.
Line 139: Define XDR before the use of abbreviation.
Lines 175 and 189: Standardize font and size.
Results:
Line 199: Review the beginning of the sentence.
Lines 235-236: Use only the previously defined abbreviation.
Line 241: Number the subitem (3.4.1).
Line 251: Number the subitem (3.4.2) and use and use italics according to taxonomic rules.
Line 259: Number the subitem (3.4.3)
Line 266: Is there a unit for these numbers?
General comments:
- Grammatical correction of minor errors in the English language is required.
- Throughout the text there are some problems with the space between words. Please check and correct.
Author Response
Dear Madam/Sir,
Thank you for your valuable comments. Enclosed is our reply and corrections according to your suggestions.
Abstract:
Q.Line 14: change to "multidrug-resistance (MDR)" or "multidrug-resistant (MDR) pathogens"
- Corrected to MDR pathogens (pathogens is added)
Q.Line 17: Define ICU and XDR in the abstract before the use of abbreviation.
A.The abbreviations are clarified: ICU is intensive care unit, XDR-extensively drug resistant
Introduction:
Q.Lines 31-34: The correlation of COVID with other illnesses was slightly vague in this sentence, being further clarified in the next sentence. I suggest readjusting this excerpt, or removing it, as the content is better described in the next sentence.
- The sentence has been removed.
Q.Lines 33 and 35: Cite the abbreviation only once.
A.The abbreviations VAP and BSI are explained when mentioned for the first time. Later, only the abbreviations are used.
Q.Line 37: Define ESKAPE before the use of abbreviation.
A.The abbreviation is explained: (Enterococcus faecium, Staphylococcus aureus, Klebsiella pneumoniae, Acinetobacter baumannii, Pseudomonas aeruginosa, and Enterobacter spp.)
Q.At the end of the introduction, mention the research objectives.
- The purpose of the study is added. The aim of the study was to analyze MDR/XDR organisms associated with COVID-19 secondary bacterial infections, their resistance determinants and molecular epidemiology.
Material and methods:
Q.Line 75: Define ARDS before the use of abbreviation.
- We have explained the abbreviation ARDS: acute respiratory distress syndrome
- Q. Lines 77-81: Describe whether samples were collected from all patients, but only 49 bacteria were isolated, or if only 49 samples were collected.
- The samples were collected from all patients, but only 49 had MDR/XDR bacteria. The other had susceptible isolates or the specimens were sterile. However, according to the second reviewer we had to separate isolates from clinical relevant specimens associated with severe infections (ETAs and BCs) from those originating from urinary tract and wound swabs or surveillance cultures (throat swabs, nasofaringeal swabs, stool, rectal swab, axilla swab). We had to put the irrelevant specimens and isolates in the supplementary material.
Q.Lines 80-81: The order must be inverted, first the full name, then, in parentheses, the abbreviation.
- Corrected: matrix assisted laser desorption ionization-time of flight mass spectrometry (MALDI-TOF MS).
Q.Lines 82-89: The statistics session must be the last of the methods session.
- A. The statistics section has been relocated to the end of the material and methods
Q.Tables 1-4: Mention in the description of the table that these values are the reference standards. In addition, it is appropriate to indicate the source of the reference used in the legend of the table (CLSI, EUCAST, etc), even if it has already been cited in the text.
- A. It is added in the table legend that MICs were interpreted according to CLSI except of colistin which was done in concordance with EUCAST. Disk.difusion test was performed and interpreted according to the CLSI guidelines.
Q.Line 123: There is no need for [CLSI] on that line. The CLSI has already been mentioned as a reference of the method previously.
- A. CLSI has been removed and the number of reference added.
Q.Line 139: Define XDR before the use of abbreviation.
- The abbreviation XDR has been explained: extensively-drug-resistant or XDR
Q.Lines 175 and 189: Standardize font and size.
- A. We have checked and the font is Times New Roman, size 12 everywhere.
Results:
Q.Line 199: Review the beginning of the sentence.
- Forty-six out of 118 patients (39%) developed serious bacterial infection (VAP or BSI or both) during stay in ICU. VAP (late onset) was diagnosed in 34 patients (29%). Twenty-five patients (21%) had BSI. Thirteen patients (11%) had simultaneously VAP and BSI. The other reviewer required to separate serious infections from urinary tract infections and wound infections. We had to put the isolates from less serious infections and colonizations in supplementary material.
Q.Lines 235-236: Use only the previously defined abbreviation.
- ETA is used instead of endotracheal aspirate as explained in the material and methods section
Q.Line 241: Number the subitem (3.4.1).
Line 251: Number the subitem (3.4.2) and use and use italics according to taxonomic rules.
Line 259: Number the subitem (3.4.3)
A.The submitems are added: 3.4.1. for Enterobacterales, 3.4.2. for A. baumannii and 3.4.3. for Gram- positive bacteria. All genus and species names are italicized.
Q.Line 266: Is there a unit for these numbers?
- There are no units for conjugation frequency. It is obtained by dividing of the number of tranconjugant cells with the number of donor cells.
General comments:
- Q. Grammatical correction of minor errors in the English language is required.
- A. Grammatical and typographical errors have been corrected.
Q.Throughout the text there are some problems with the space between words. Please check and correct.
- A. superfluous spaces have been deleated.
Reviewer 2 Report
Bedenic et al descibed the epidemiology of bacterial resistance among patients admitted for severe Covid-19 infection in a national reference unit. Although extensive bacteriological workup had been performed, the manuscript cannot be published in Pathogens without subsequent modifications.
Major comment
+ Results The reader cannot understand how many Ventilator associated pneumonia (VAp) are clearly due to Acinetobacter baumanii or MDR Enterobacteriales. The results should first give the clinical data of the whole cohort. The sole analysis of mortality is not a good reflect of prognosis of bacterial resistance because mortality could be due to Covid-19 itself. The authors should tell how many VAP, how many bacteriaemia occurred in these patients and should focus on these VAP/Bacteraemia. How many VAP and bacteraemia were due to MDR/XDR pathogens? The authors should analyse by logistic regression the independent impact of the occurrence of VAP/Bacteremia (composite factor) onto mortality as well as the independent impact of MDR/XDR pathogens.
+Results : Then, the authors should focus their bacteriological analysis on the sole strains involved in truly infective samples (BronchioAlveolar Lavage or Endotrocheal Aspirate samples and bacteriaemia). It would be easier to understand for readership of Pathogens. The detail of MDR/XDR among others samples should be given in percentage (others samples and percentage, then detail of the others samples). Subsequent analysis of these potentially non infective samples should be given as supplementary material.
+Discussion should be rewritten in line with these new results.
Minor Comment
+ introduction
- Line 37 are
- Where is the objective of the study in the introduction part?
+ material
- Line 69 study period
- Line 89 considered as stastically significant
- Line 133 please explain “heavy”. How many Mc farland units?
- Line 148 “was also determined” instead of “was determined as well”
- Line 180 reference order is incorrect [50] between [31] and [33]
+ results
- Statistical analysis should not be a subheading in the results part
- Table 1 and 2 did not provide data about specimen
- Line 198 – 199 consider rephrasing the first paragraph: first how many patients first,then how many infections and lastly how many VAP? Please also consider to first give the whole clinical data before the number of strains (see above)
+ conclusion
- Line 395 “spread” instead of “spreading”
Author Response
Dear Madam/Sir,
Thank you for your valuable comments. Enclosed is our reply and the list of amendments and corrections.
Bedenic et al descibed the epidemiology of bacterial resistance among patients admitted for severe Covid-19 infection in a national reference unit. Although extensive bacteriological workup had been performed, the manuscript cannot be published in Pathogens without subsequent modifications.
Major comment
- Results The reader cannot understand how many Ventilator associated pneumonia (VAp) are clearly due to Acinetobacter baumanii or MDR Enterobacteriales. The results should first give the clinical data of the whole cohort. The sole analysis of mortality is not a good reflect of prognosis of bacterial resistance because mortality could be due to COVID-19 itself. The authors should tell how many VAP, how many bacteriaemia occurred in these patients and should focus on these VAP/Bacteraemia. How many VAP and bacteraemia were due to MDR/XDR pathogens? The authors should analyse by logistic regression the independent impact of the occurrence of VAP/Bacteremia (composite factor) onto mortality as well as the independent impact of MDR/XDR pathogens.
A.Forty-six out of 118 patients (39%) developed serious bacterial infection (VAP or BSI or both) during stay in ICU. VAP (late onset) was diagnosed in 33 patients (28%). Twenty-seven patients (23%) had BSI. Fourteen patients (12%) had simultaneously VAP and BSI. All patients with serious infections had elevated white blood cells count, CRP and X ray result consistent with severe bilateral pneumonia. For the clinicians it was difficult to distinguish if it the deterioration was due to secondary bacterial infection or progression of COVID-19, but most probably due to the combination of ARDS and secondary bacterial infections, leading to multiorgan failure (MOF). The mortality rate due to bacterial infection or combination of ARDS with bacterial superinfection was 33% (40/118). The lethal outcome was recorded in 33/40 patients (80%) having serious bacterial infection due to MDR or XDR organisms and in seven out of eight patients (87%) with susceptible organisms associated with BSI. Thus, 40 death cases (48%) were attributed to bacterial infection, complicating COVID-19 disease, while the remaining 43 patients (52%) died from the progression of COVID-19 disease. Less serious infections diagnosed in patients were urinary tract infections (UTI) diagnosed in eight (6.7%) patients and wound infections identified in three (2.5%) of the patients. Age and gender were not found to be a risk factor for acquisition of MDR or XDR isolates with P values 0.7 and 0.4, respectively. Comorbidities such as diabetes mellitus, asthma, fatness, polyarthritis, chronic kidney failure and malignant diseases were not identified as risk factors for infections with resistant isolates with P values of 0,4, 0,11, 1.00, 0,61, 0.73 and 0.65, respectively. Significant P value was only obtained for cardiovascular diseases (P=0,03). Logistic regression showed strong positive relationship between the presence of serious infection and the mortality (P=0.001, odds ratio 4.341 with 95% CI 1.627 to 11.581, AUC 0.653 with 95% CI 0.559 to 0.738). On the other hand, it showed no relationship between the presence of the XDR/MDR bacterial strain as the causative agent of the serious infection and mortality (P=0.960, odds ratio 1.061 with 95% CI 0.107 to 10.544, AUC 0.504 with 95% CI 0.353 to 0.655).
Q.Results : Then, the authors should focus their bacteriological analysis on the sole strains involved in truly infective samples (Bronchioalveolar Lavage or Endotrocheal Aspirate samples and bacteriaemia). It would be easier to understand for readership of Pathogens. The detail of MDR/XDR among others samples should be given in percentage (others samples and percentage, then detail of the others samples). Subsequent analysis of these potentially non infective samples should be given as supplementary material.
- We have removed from the results and tables the data pertaining to MDR bacteria from other non-relevant specimens such as urine, wound swabs or surveillance cultures. More clinical data were added in the results section and material and methods. The protocol for VAP and BSI diagnostics and the sampling procedures are added.
Q.Discussion should be rewritten in line with these new results.
- The discussion is rewritten according to the new results. The section on OXA-48 producing K. pneumoniae has been removed since they were all isolates from urine or surveillance cultures. The section on MRSA has been added. Moreover, we have explained that bacterial infection itself was related to the poor outcome. However, the presence of resistant bacteria did not increase the mortality rate.
Minor Comment
introduction
- Q. Line 37 are
- A. corrected
- Q. Where is the objective of the study in the introduction part?
A.The objective of the study has been added: The aim of the study was to analyze MDR/XDR organisms associated with COVID-19 secondary bacterial infections, their resistance determinants and molecular epidemiology.
material
Q.Line 69 study period
- A. We have added the study period: during November and December 2020,
- Line 89 considered as statistically significant
A.Corrected: as is added
- Line 133 please explain “heavy”. How many Mc farland units?
- The suspension was adjusted to Mc Farland 0.5 corresponding to 108 CFU/ml.
- Q. Line 148 “was also determined” instead of “was determined as well”
- A. corrected: Cotransfer of resistance to gentamicin, tetracycline, sulfamethoxazole/trimethoprim,chloramphenicol and ciprofloxacin was also determined
- Line 180 reference order is incorrect [50] between [31] and [33]
- A. corrected: Carattoli et al is number 32
results
Q.Statistical analysis should not be a subheading in the results part
- A. corrected. The statistical analysis is merged with the patient’s section.
Q.Table 1 and 2 did not provide data about specimen
- A. There was initially column with the specimens but I had to delete it because the tables were too wide. Initial tables were in landscape format but I had to convert them to portrait format because the whole manuscript must fit in the form provided by the journal. Gram-positive bacteria have less antibiotics in the panel for testing and thus there is no need for so many columns.
- Q. Line 198 – 199 consider rephrasing the first paragraph: first how many patients first, then how many infections and lastly how many VAP? Please also consider to first give the whole clinical data before the number of strains (see above)
- A. The paragraph has been rephrased according to the suggestions.
conclusion
Q.Line 395 “spread” instead of “spreading”
A.corrected: spread instead of spreading
Submission Date
07 December 2022
Reviewer 3 Report
Nice study, please see my comments below:
Comments
Line 17: Please mention the full form of MDR and XDR. Also, please try to mention the MDR and XDR results separately.
Line 17-19: Please provide this information into a single sentence.
Line 21-22: Please provide full name of the organisms here.
Line 34-35: You have already used the full form of VAP. No need to use it again, just use VAP here.
Line 37: Please provide the full form of ESKAPE (mention all the organisms’ name here)
Line 39: “…..important multidrug-resistant bacteria (MDR) causing…” should be “…..important multidrug-resistant (MDR) bacteria causing…”
Line 42: “Acinetobacter baumannii (CRAB)..” should be “carbapenem-resistant Acinetobacter baumannii(CRAB)..”
Line 42: Please remove “and”
Line 39-44: Please provide a reference for this statement.
Line 75: Please provide the full form of “ARDS” here
Line 81: Please an appropriate reference or briefly describe the procedure of MALDI-TOF here
Line 88: Chi-square or chi-square for relatedness? Please check it.
Line 89: “P value”, here P should be italic. Please correct it throughout the manuscript.
Line 196: This manuscript analyses? It should be “This study analyzed”
Line 229: P value 6.61? Please correct it.
Line 234-236: Please remove this sentence. You have already mentioned it in the materials and methods section.
Table 3: Are isolates 1 to 5 MDR in nature? According to your table, are they resistant to three or more antimicrobial classes? Please check it.
Line 255: Bracket error
Line 261: According to your Table 4, four isolates were resistant to ciprofloxacin. Please check it again and correct it.
Author Response
Dear Madam/Sir,
Thank you for your valuable comments. Enclosed is our reply and amendments according to your suggestions
Q.Line 17: Please mention the full form of MDR and XDR. Also, please try to mention the MDR and XDR results separately.
- We have explained the abbreviations MDR and XDR and shown the two groups separately. However, for the purpose of statistical analysis they were merged together because there were only a few MDR isolates associated with severe infections (VAP and BSI). The majority of the resistant bacteria were A. baumannii isolates with XDR phenotype. The second reviewer required to exclude resistant bacteria from non severe infections such as urinary tract infections and colonizations. Thus, there was only one MDR E. coli and five MRSA isolates, left. All other were XDR. A baumannii.
Q.Line 17-19: Please provide this information into a single sentence.
- A. The information in lines 17-19 is already in one sentence.
- Line 21-22: Please provide full name of the organisms here.
- The genus and species names of bacteria are provided in full names when mentioned for the first time. Later, they are abbreviated.
- Q. Line 34-35: You have already used the full form of VAP. No need to use it again, just use VAP here.
- The full form of VAP was mentioned for the first time in the first line of the introduction section. Later only the abbreviation was used. The first sentence in the first version of the manuscript has been removed as required by the first reviewer.
Q.Line 37: Please provide the full form of ESKAPE (mention all the organisms’ name here)
- The abbreviations ESKAPE is explained: Enterococcus faecium, Staphylococcus aureus, Klebsiella pneumoniae, Acinetobacter baumannii, Pseudomonas aeruginosa, and Enterobacter spp.
Q.Line 39: “…..important multidrug-resistant bacteria (MDR) causing…” should be “…..important multidrug-resistant (MDR) bacteria causing…”
- A. Corrected: Important multidrug-resistant bacteria (MDR) causing
- Line 42: “Acinetobacter baumannii(CRAB).” should be “carbapenem-resistant Acinetobacter baumannii (CRAB).”
- A. Corrected: carbapenem-resistant Acinetobacter baumannii (CRAB)
Q.Line 42: Please remove “and”
- A. and has been removed
Q.Line 39-44: Please provide a reference for this statement.
- A. Reference for MRSA is added: Deurenberg RH, Sobbering EE. The evolution of Staphylococus aureus. Infection, Genetics and Evolution 2008; 8: 747-763; doi: https://doi.org/10.1016/j.meegid.2008.07.00
Q.Line 75: Please provide the full form of “ARDS” here
A.Provided: acute respiratory distress syndroma (ARDS)
Q.Line 81: Please an appropriate reference or briefly describe the procedure of MALDI-TOF here
- The reference has been added: Dingle, T.C; Butler-Wu, S.M. MALDI-TOF mass spectrometry for microorganism identification. Clin Lab Med. 2013, 3, 589-609. doi: 10.1016/j.cll.2013.03.001.
Description of the method: One or two colonies of each isolate were directly spotted on the manufacturer’s proprietary sample plates following the manufacturer`s protocols and recommendations. A 1 µL volume of CHCA matrix solution (α-cyano-4-hydroxycinnamic acid; bioMérieux Inc.) was then applied to each sample and air-dried for 5 min at room temperature for crystallization. For species identification of each isolate, a total of four spots were analyzed on the VITEK MS system. The MALDI-TOF MS instrument used in this study was equipped with a 337 nm-fixed focus nitrogen laser of 50Hz frequency, the software program was the VITEK MS IVD analysis software version 3.2. Escherichia coli ATCC 8739 was used as positive control.
Q.Line 88: Chi-square or chi-square for relatedness? Please check it.
A.Chi square
Q.Line 89: “P value”, here P should be italic. Please correct it throughout the manuscript.
- A. Corrected: P is italicized throughout the text
Q.Line 196: This manuscript analyses? It should be “This study analyzed”
- A. The sentence has been removed from the text. The result section is in the past tense.
- Q. Line 229: Pvalue 6.61? Please correct it.
A.This was an error. P. value is 0,61
Q.Line 234-236: Please remove this sentence. You have already mentioned it in the materials and methods section.
- A. The sentence has been removed.
- Q. Table 3: Are isolates 1 to 5 MDR in nature? According to your table, are they resistant to three or more antimicrobial classes? Please check it.
- The isolates were resistant to microlides and clindamycin as well, but the results were not included in the table because these antibiotics are bacteriostatic and not used in seriously ill patients. According to your suggestion we have added susceptibility results for these antibiotic groups. All five isolates exhibited resistance to at least one antibiotic in three different classes.
Q.Line 255: Bracket error
- A. Bracket error has been corrected: (n=25)
Q.Line 261: According to your Table 4, four isolates were resistant to ciprofloxacin. Please check it again and correct it.
- A. Yes, there were three ciprofloxacin resistant isolates instead of two in the first version of the manuscript, but we had to remove one strain as it was isolated from the surveillance culture. The second reviewer required to keep only clinically relevant isolates (ETA and BC).
Round 2
Reviewer 2 Report
The authors have taken into account my previous remarks.
I would suggest these final minor comments :
line 214 methicillin susceptible
line 308 no correlation
discussion the authors should state the limit that their study might not have the power to show an independant association between MDR XDR bacterial phenotypes and mortality
Author Response
Dear Madam/Sir,
Thank you for your valuable comments. Enclosed is our reply
The authors have taken into account my previous remarks.
I would suggest these final minor comments :
- line 214 methicillin susceptible
- corrected: methicillin susceptible Staphylococcus spp
Q.line 308 no correlation
- corrected: no correlation instead of to correlation. This was a typographical error.
- discussion the authors should state the limit that their study might not have the power to show an independant association between MDR XDR bacterial phenotypes and mortality
- We have explained that the limit of the present study is that it might not have the have the power to show an independant association between MDR and XDR bacterial phenotypes and mortality.